# In Vivo Preclinical Assessment of β-Amyloid–Affine [^11^C]C-PIB Accumulation in Aluminium-Induced Alzheimer’s Disease-Resembling Hypercholesterinaemic Rat Model

**DOI:** 10.3390/ijms232213950

**Published:** 2022-11-12

**Authors:** Zita Képes, Alexandra Barkóczi, Judit P. Szabó, Ibolya Kálmán-Szabó, Viktória Arató, István Jószai, Ádám Deák, István Kertész, István Hajdu, György Trencsényi

**Affiliations:** 1Division of Nuclear Medicine and Translational Imaging, Department of Medical Imaging, Faculty of Medicine, University of Debrecen, Nagyerdei St. 98, H-4032 Debrecen, Hungary; 2Department of Urology, Faculty of Medicine, University of Debrecen, Nagyerdei St. 98, H-4032 Debrecen, Hungary; 3Doctoral School of Clinical Medicine, Faculty of Medicine, University of Debrecen, Nagyerdei St. 98, H-4032 Debrecen, Hungary; 4Gyula Petrányi Doctoral School of Clinical Immunology and Allergology, Faculty of Medicine, University of Debrecen, Nagyerdei St. 98, H-4032 Debrecen, Hungary; 5Department of Operative Techniques and Surgical Research, Faculty of Medicine, University of Debrecen, Nagyerdei St. 98, H-4032 Debrecen, Hungary

**Keywords:** Alzheimer’s disease (AD), aluminum (Al), [^11^C]C-Pittsburgh compound B ([^11^C]C PIB), hypercholesterinaemia, positron emission tomography (PET), standardised uptake value (SUV)

## Abstract

Aluminum (Al) excess and hypercholesterinaemia are established risks of Alzheimer’s disease (AD). The aim of this study was to establish an AD-resembling hypercholesterinaemic animal model—with the involvement of 8 week and 48 week-old Fischer-344 rats—by Al administration for the safe and rapid verification of β-amyloid-targeted positron emission tomography (PET) radiopharmaceuticals. Measurement of lipid parameters and β-amyloid–affine [^11^C]C-Pittsburgh Compound B ([^11^C]C-PIB) PET examinations were performed. Compared with the control, the significantly elevated cholesterol and LDL levels of the rats receiving the cholesterol-rich diet support the development of hypercholesterinaemia (*p* ≤ 0.01). In the older cohort, a notably increased age-related radiopharmaceutical accumulation was registered compared to in the young (*p* ≤ 0.05; *p* ≤ 0.01). A monotherapy-induced slight elevation of mean standardised uptake values (SUV_mean_) was statistically not significant; however, adult rats administered a combined diet expressed remarkable SUV_mean_ increment compared to the adult control (SUV_mean_: from 0.78 ± 0.16 to 1.99 ± 0.28). One and two months after restoration to normal diet, the cerebral [^11^C]C-PIB accumulation of AD-mimicking animals decreased by half and a third, respectively, to the baseline value. The proposed in vivo Al-induced AD-resembling animal system seems to be adequate for the understanding of AD neuropathology and future drug testing and radiopharmaceutical development.

## 1. Introduction

Provided the immense burden that cognitive disorders such as Alzheimer’s disease (AD) and associated health ramifications impose on societies, in-depth understanding of the molecular neuropathological continuum of AD-derived brain alterations is one of the priorities of todays’ science. Identification of early stages of AD is gaining increasing attention since timely therapeutic intervention is regarded the most effective in the preclinical forms of the disease [1].

Nuclear medical preclinical animal models recapitulating AD appear to shed light on the yet unresolved mechanisms behind it, opening the way towards the discovery of novel diagnostic biomarkers and therapeutic targets. Pioneering transgenic rat and mice models such as *APP/PS1*, *PS2APP* (*PS2N141I* × *APPswe*), and *3*×*Tg-AD* mimicking AD development were hailed as landmarks in the comprehension of disease pathology [2,3,4,5]. Greater body and brain dimensions connected to a less frequent incidence of partial volume effects in preclinical models composed of rats gained superiority over translational mice systems in brain imaging [6].

One transgenic rat model—termed as bigenic *TgF344-AD*—offers appropriate scenario for disease characterisation as it is featured with AD-associated intracellular β-amyloid (iAβ), β-amyloid-40 (Aβ-40), and β-amyloid-42 (Aβ-42) peptides and Aβ plaque depositions [7,8]. Moreover, it exhibits Gallyas-positive structures with *p-tau* as well that are similar to those of the human neurofibrillary tangles (NFTs)—one of the hallmarks of AD-neuropathology [7,8]. Further, *SHR24* or *hTau-40/P301L* rats expressing *tau* have also made a leap forward the better understanding of neurodegeneration [9,10]. Therefore, these AD-like translational models with *tau* or Aβ overexpression might enable the assessment of brain changes prior to the appearance of manifest clinical signs of cognitive deterioration. This could ultimately lead to therapeutic advances which may delay or even prevent disease formation.

Although transgenic types of models are widespread in the preclinical study of AD, the establishment of such model systems bears spiralling costs aggravated by the difficulties of the special circumstances required for animal maintenance. Spontaneous AD-like cat or monkey models may serve as potential alternatives for preclinical research; however, these generally ensure post-mortem analysis only with the enrollment of a relatively insufficient number of experimental animals [11,12]. Further, alternative models involving lower living organisms such as *Caenorhabditis elegans* could be applied solely in genetic studies [13,14,15]. Given the cost-effectiveness, and the rapid disease development achieved within the framework of chemically induced AD small animal systems, these seem to be the mainstream tools for AD research.

Positron emission tomography (PET) applying [^18^F]-2-fluoro-deoxy-D-glucose ([^18^F]F-FDG) is supposed to be of significant potential to decipher subtle brain metabolic alterations that precede the appearance of morphological cerebral changes. Based on research data [^18^F]F-FDG brain hypometabolism is a promising biomarker of AD [16]. According to prior studies reporting decreased [^18^F]F-FDG accumulation of the amygdala, the entorhinal cortex, and the hippocampus, [^18^F]F-FDG PET appeared to be feasible in the identification of AD-associated brain metabolic patterns in a Wistar male rat model of chronic cerebral hypoperfusion (CCH) [17]. In another study, the development of neurodegeneration was strengthened by—amongst others—the serial registration of [^18^F]F-FDG hypometabolism of the anterodorsal hippocampus in rats with bilateral common carotid artery ligation-based CCH [18]. Although [^18^F]F-FDG is valued as the gold-standard PET radiopharmaceutical in the evaluation of AD-linked brain glucose consumption and glucose metabolic changes, its application in neurodegenerative imaging has been hampered by its low specificity in terms of differentiating AD from other forms of dementias [19,20].

A novel amyloid PET technique applying the thioflavin T derivate ^11^[C]C-Pittsburgh compound B (^11^[C]C-PiB) excels in the detection of AD pathognomonic insoluble extracellular Aβ plaques [21,22]. ^11^[C]C-PiB PET examinations revealed age-dependent enhanced ^11^[C]C-PiB accumulation in particular cerebral areas of the *APP/Tau* rat model (male *McGill-R-Thy1-APP* and female *hTau-40/P301L* transgenic rats) compared to the control cohort [23].

Another aspect is that aluminium (Al) consumption and the development of neurodegenerative disorders such as AD are closely connected [24]. Besides facilitating *tau* phosphorylation and subsequent NFT formation, Al may influence Aβ kinetics [25,26,27]. The neurotoxic characteristics of Al were first revealed in 1897 in preclinical small animals [28]. Later in 1989, Nertholf and colleagues reported NFT accumulation in the oculomotor nucleus of New Zealand white male rabbits i.v. administered with Al maltol [29]. Moreover, vascular alterations developed on the basis of high cholesterol levels may elevate the likelihood of the occurrence of AD [30]. Ground-breaking evidence from 1994 explored for the first time immunolabelled AD-resembling Aβ deposition in rabbits’ brains kept on cholesterol-rich diet [31]. The association between hypercholesterinaemia and heightened risk for the development of sporadic AD was further strengthened by the observation of AD-mimicking characteristics in a preclinical study involving a high-fat cholesterol diet (HFCD)-fed triple transgenic mouse model of AD (*3xTgAD*) and non-transgenic control mice [30].

Initiated by the above detailed facts, in the present study, we aimed at proposing a novel chemically induced AD-like rat model featured with rapidly accumulating Aβ-plaques that may be suitable for the safe and rapid verification of β-amyloid-based PET radiopharmaceuticals.

## 2. Results

### 2.1. Results of Blood Test Analysis

To assess the effect of a butter and cholesterol-rich (BCR) diet on serum lipid levels, cholesterol and low-density lipoprotein (LDL) values were measured after the one-month long administration of a high-cholesterol diet (*ad libitum*) to the experimental animals. Laboratory data led to the recognition that the LDL (approximately 4 mmol/L) and total cholesterol (approximately 6 mmol/L) levels of the animals receiving the cholesterol-rich diet were significantly elevated compared to those of the untreated, control animals (0.2 mmol/L and 1.3 mmol/L, for the LDL and the total cholesterol levels; respectively, *p* < 0.01). Laboratory data are presented in Figure 1.

### 2.2. In Vivo Positron Emission Tomography (PET) Imaging Results after One Month of Combined (Cholesterol and Al_2_(SO_4_)_3_) Treatment

Eight week-old (young) and forty-eight week-old (elderly) male Fischer-344 rats were employed in our series of experiments. Comparing the transaxial PET images of the control old and young rats, clearly identifiable increased [^11^C]C-PIB radiopharmaceutical accumulation could be encountered on the images of the elderly in comparison with the cerebral tracer uptake of the young animals (demonstrated in Figure 2A, left; and Figure 2C, left). Based on this finding we assume that age difference alone triggers Aβ plaque development. The extent of the increment of [^11^C]C-PIB uptake in the treated (cholesterol and Al_2_(SO_4_)_3_) groups of both the old and the young rats is conspicuous (demonstrated in Figure 2A, right; and Figure 2C, right). Mean standardised uptake values (SUV_mean_) have risen from 0.78 ± 0.16 to 1.99 ± 0.28 and from 0.45 ± 0.11 to 0.99 ± 0.18 in the group of the elderly and the young rats receiving combined (cholesterol and Al_2_(SO_4_)_3_) treatment, respectively (Figure 2B,D). Since the qualitative findings were more obvious in the cohort of the elderly, we continued the research with the exclusive involvement of only this group.

During the rest of the present study, we also aimed at comparing the SUV results of the animals receiving monotherapy (cholesterol or Al_2_(SO_4_)_3_) and combined treatment (cholesterol and Al_2_(SO_4_)_3_) to each other and to the control group (normal rodent feed with normal drinking water) as well (Figure 3). Although a monotherapy-induced slight elevation of SUV_mean_ values was defined, no statistically significant difference was pinpointed regarding the comparison of the radiopharmaceutical uptake of the rats kept on monotherapy and that of the control group (presented in Figure 3B). However, in case of the rats administered a combined (cholesterol rich and Al_2_(SO_4_)_3_) diet, a significant SUV_mean_ increase was depicted (demonstrated in Figure 3B).

### 2.3. Results of PET Examinations One and Two Months after Restoration to Normal Diet and Water

Brain PET examinations were performed one and two months after the reintroduction of standard diet and tap water. As previously mentioned, considerably elevated SUV_mean_ values were registered following a month-long combined (cholesterol rich and Al_2_(SO_4_)_3_) treatment. Then, the experimental animals were restored to the same standard rodent chow and water that were administered to the rats of the control group. We surprisingly experienced that following a month-long normal diet, cerebral [^11^C]C-PIB accumulation (SUV_mean_: 0.99 ± 0.21; SUV_max_: 4.23 ± 1.36) significantly decreased by half of the value measured after a month of combined treatment. Additionally, two months post-restoration, the tracer uptake decreased to a third (SUV_mean_: 0.55 ± 0.27; SUV_max_: 3.35 ± 1.47, as presented in Figure 4).

## 3. Discussion

In view of the rising worldwide prevalence of AD, there is a growing demand for the introduction of novel drug candidates and reliable diagnostic methods to combat the pertinent global threat caused by neurodegenerative disorders. Elevation of serum cholesterol level and an excessive amount of Al are regarded as risk factors of AD [32,33]. Formerly, preclinical research into the continuum of cognitive impairment has been hampered by the relative paucity of adequate in vivo small animal systems. Translational nuclear medical models may grant promising scenarios for the mechanistic understanding of AD neuropathology. Since we hypothesise that preclinical systems may contribute to the investigation of new reliable therapeutic targets and diagnostic approaches, we intended to propose an experimental rat model that enables future drug trials and radiopharmaceutical development in different groups of AD-like rats fed with Al and/or cholesterol rich diet.

Within the framework of our experiments, chemically induced small animal systems resembling AD were in vivo examined applying PET. [^11^C]C-PIB produced in the radiochemical laboratory of the Department of Nuclear Medicine and Translational Imaging, University of Debrecen was the central radiotracer of our series of experiments. Since [^11^C]C-PIB specifically binds to the well-defined lesions of AD—the β-amyloid plaques—such AD-mimicking preclinical models allows for the tracking of the development of these pathological hallmarks.

Ample evidence reports on the establishment of different cholesterol-rich diet triggered animal models of AD. Sparks et al. stated that high cholesterol diet-induced accumulation of β-amyloid plaques could be achieved in New Zealand white rabbits [34]. Provided the expenses and the special circumstances necessary for rabbit maintenance, interest in the application of smaller rodents has come to focus. Since guinea pigs are not transgenic animals and their LDL molecule is comparable to that of the human, making them an appropriate representative of human lipoprotein and cholesterol metabolism, they serve as potential alternatives for the development of AD-like preclinical systems [35]. Although, similarly to rabbits, special maintenance requirements increase the costs of the proposal of such translational models.

Since we aimed at establishing a cost-effective AD animal model, due to financial aspects we excluded the application of both rabbits and guinea pigs.

Prior literature data are available on high cholesterol-induced AD animal models in the case of small rodents, including rats. Cognitive deterioration in association with diets rich in cholesterol and saturated fatty acids with various composition was assessed in the case of Long Evans rats [36]. AD-related cognitive dysfunction was clearly verified by their learning and memory impairments [36]. In a study of Kirsch and co-workers, with the enrolment of 8-weeks-old albino Wistar rats fed with 2% cholesterol-containing rodent chow for 6 months, they pinpointed notable difference between the serum cholesterol levels of the treated animals and the rats on normal rodent chow [37]. In accordance with their results, we also experienced considerable difference between the serum cholesterol values of the control animals and those fed with the special cholesterol enriched diet. This finding strengthens the finding that we successfully managed to achieve diet-based hypercholesterinaemia in the group of animals receiving the special feed with high cholesterol content.

As the positron emitter ^11^C-labelled PIB exerts specific affinity to the AD pathognomonic Aβ plaques, it facilitates the PET diagnostics of AD. Based on the amyloid cascade hypothesis, the cerebral accumulation of Aβ plaques is widely accepted as a key element of manifest disease development [38]. Further, due to the capability of PIB to cross the blood–brain barrier it could be used as a carrier to ease the central transport of those molecules that otherwise would not be able to enter the brain. Previous research studies—highlighting a PIB derivate as a potential drug carrier—report on the Aβ specificity of PIB molecule as well. Therefore, the application of PIB not only provides opportunity for the diagnostics of different neurological disorders, such as AD, but also for the widening of the existing therapeutic armamentarium of these diseases [39].

In the present study we observed enhanced brain radiotracer uptake of the old rats in comparison with the younger animals. The natural course of aging and age-related increased Aβ plaque deposition could be a possible explanation behind this. In a recent review, Poon and colleagues also reported about the increment of AD-linked Aβ plaques dependent upon aging in mice with APP mutations of familial AD [40].

However, in both the old and young group of rats administered with combined diet (Al_2_(SO_4_)_3_ and cholesterol), significantly elevated tracer uptake was depicted compared to the control group during the analyses of the PET scans. We presuppose that the vulnerability of the brain of the elderly animals could possibly undermine why the diet-induced brain changes led to more visible SUV elevation in the older cohort compared to the young. We further assume that both Al and cholesterol—though via not yet fully elucidated mechanisms—trigger the development of age-related cerebral alterations that eventually result in the formation of AD-like perturbations such as Aβ plaque accumulation with subsequent tracer uptake increment.

In addition, the application of monotherapy (Al_2_(SO_4_)_3_ or cholesterol) did not generate notable difference between the PIB accumulation of the monotreated groups and the healthy control cohort. Prior histopathological data also revealed alarming cerebral cortical impairment in male albino Wistar rats administered with sublethal daily doses of AlCl_3_ [41]. We suggest that prolonged treatment duration may induce meaningful brain changes with more pronounced tracer uptake elevation. One-to-two months follow-up of animals restored to a normal diet—formerly kept on a month-long combined (Al_2_(SO_4_)_3_ and cholesterol) diet—was executed as well. We surprisingly experienced that following the one-month restoration of normal diet (standard chow and water), brain [^11^C]C-PIB accumulation decreased by half of the value measured after a month of combined (Al_2_(SO_4_)_3_ and cholesterol) treatment, whereas two months after the reintroduction to normal chow and water, SUV values decreased to a third. A kind of spontaneous regression of Aβ plaques may possibly underpin the reduction of [^11^C]C-PIB uptake after restoration to standard diet. This emphasises the reversible characteristic of plaque accumulation that may have pivotal human implications regarding timely diagnostic assessment and early therapeutic or preventive intervention. Although the entire molecular mechanism regarding the removal of Aβ plaques remains to be elucidated, widespread interest in amyloid elimination has spawned an immense number of research studies dealing with Aβ degradation. Formation of Aβ plaques takes place partly in the endosomes. A substantial portion of these amyloid plaques are degraded by endothelin-converting enzymes (ECE), undefined proteases and lysosomal cathepsins [42]. Further, either by the Golgi compartment or from the endosome itself, recycling vesicles filled with amyloid could be transported to the surface of the cell. Then, ubiquitin-proteasome system-based (UPS) degradation of the plaques or clearance by various pathways including neprylisin (NEP), insulin-degrading enzyme (IDE), and matrix metalloproteinase-9 (MMP-9) might arise [42]. In addition, cellular proteolysis is regulated mainly by proteasomes and lysosomes [42]. Besides these, Ca^2+^-triggered cysteine proteases—calpains—and cysteine-dependent aspartate-directed caspases have a pivotal role in the generation of cell injury and death [42,43]. Based on prior research data, amyloid plaques could be encountered in lymph nodes as well; however, the lymphatic elimination of Aβ plaques seems to be underappreciated [42,44]. In this respect, dendritic and microglial cells are reported to take part in amyloid clearance and degradation [42]. The neuronal recovery-based reversibility of plaque formation was strengthened in a study conducted by Tanokashira et al. [45]. Moreover, by stating the efficacy of anti-Aβ antibodies in the recovery of neuritic dystrophy developed based on plaque deposition, Brendza and co-workers also supported the changeability of Aβ-derived neuronal toxicity applying APP transgenic mouse model [46].

According to a study from 2016, non-enzymatic mechanisms, such as modifications in the N-terminal region might explain the degradation of the plaques [47]. However, the exact mechanism behind is not yet entirely uncovered, and long-term unbiased future studies are required to fill the gap.

There are some limitations of the present research that need to be addressed. Since the major aim of the current study was to investigate the connection between Al exposure, a cholesterol-rich diet and AD, as well as the development of diet-induced AD-resembling cerebral impairments, neither AD-like clinical nor neuropathological features (cognitive impairment, neuronal loss etc.) of the experimental animals was assessed. The appearance of AD-related pathology was proven solely by the [^11^C]C-PIB PET-based identification of amyloid plaques. Therefore, according to our results, the manifest occurrence of AD in the involved small animals could not be stated for sure. Future studies supplemented with neurohistopathological examinations and cognitive tests designed for preclinical experimental animals are warranted to definitely support AD manifestations in such model systems.

## 4. Materials and Methods

### 4.1. Experimental Animals

To perform our study, 8-week-old (young) and 48-week-old (elderly) male Fischer-344 rats (n = 6) were applied. During the experiments the animals were kept in a conventional animal house (University of Debrecen, Faculty of Medicine, Division of Nuclear Medicine and Translational Imaging) where the temperature and the humidity were 26 ± 2 °C and 50 ± 5%, respectively. Artificial lighting was provided in automatically controlled 12 h cycles. The rats were fed *ad libitum* with semi-synthetic feed (VRF1 rodent feed; Charles River Ltd., Gödöllő, Hungary), and depending upon the treatment, they were kept on tap water *ad libitum* and drinking water containing Al_2_(SO_4_)_3_. The experiments were performed in compliance with the European Union rules and regulations with the permission of the Ethics Committee for Animal Experimentation of the University of Debrecen. (Number of permission: 21/2017/DEMÁB; 3-1/2014/DEMÁB). Figure 5 shows the detailed study design.

### 4.2. Chemical Induction of the Experimental Animals

We executed our experiments applying 8-week-old (young) and 48-week-old (elderly) male F-344 rats (n = 4/group). During the research both the old and the young animals were divided into the following subgroups:Group 1 diet rich in butter and cholesterol and normal tap water *ad libitum.*Group 2 normal rodent chow and drinking water containing 200 mg/L Al_2_(SO_4_)_3_ *ad libitum*.Group 3 diet rich in butter and cholesterol and drinking water containing 200 mg/L Al_2_(SO_4_)_3_
*ad libitum*.Group 4 normal rodent chow and normal tap water *ad libitum* (control group).

The subsequent table demonstrates the detailed composition of the applied normal, VRF1 rodent chow (Table 1).

A special mixture was prepared for the animals fed with the diet rich in butter and cholesterol. Sixty per cent of this specialised chow contained the above mentioned standard VRF1 feed, while for the remaining 40% the following ingredients were added (as shown in Table 2).

Casein was added to the feed to supplement the amino acid need of the experimental animals. Although the cholesterol-rich diet itself leads to plasma cholesterol level elevation, adding butter to it—containing additional cholesterol and saturated fatty acids—results in the further increase of cholesterol values [48]. Sodium-cholate was required to enhance the gastrointestinal absorption of cholesterol, while methyl–thiouracil was added to reduce cholesterol levels, lowering thyroid hormone secretion through the inhibition of thyroid peroxidase. Given the high sodium and low potassium content of Sós’ SM8 Salt Mixture, it was also added to elevate the blood pressure of the animals [49]. After the four-week-long treatment (diet and/or drinking water *ad libitum*) the experimental groups were continued to be kept on normal rodent chow and drinking water *ad libitum*.

### 4.3. PET Examinations

In vivo small animal PET examinations were carried out in the University of Debrecen, Faculty of Medicine, Division of Nuclear Medicine and Translational Imaging with the application of a MiniPET scanner. The male Fischer-344 rats (n = 16) were anaesthetised with an inhalation machine designed for laboratory rodents (induction: 3% Forane, sustenance: 1.5% Forane + 0.4 L/min O_2_ + 1.2 L/min N_2_O). Then, the rats were intravenously (i.v.) administrated with β-amyloid plaque–affine [^11^C]C-PIB (12.2 ± 1.4 MBq). The 30-min long incubation period was followed by the acquisition of 20-min static PET scans on the brain of the rats under inhalation anaesthesia prior to the start of the diet, after the one-month long treatment, and one, and two months after the restoration of normal chow and water, applying the small animal PET scanner.

### 4.4. Analyses of Serum Blood Samples

The lipid profile of the experimental animals was determined from *i.v.* taken serum samples. For the measurement of total cholesterol and LDL cholesterol, the methods of Allain et al. (1974) and Sugiuchi et al. (1998) were utilised, respectively, on Roche/Hitachi cobas C systems [50,51]. The lipid fractions were expressed in mmol/L.

### 4.5. Statistical Analyses

Significance was determined by the Student’s two-tailed *t*-test, two-way ANOVA, and the Mann–Whitney rank-sum tests. The significance level was set at *p* ≤ 0.05 unless otherwise indicated. A commercial software package—named MedCalc 18.5 (MedCalc Software, Mariakerke, Belgium)—was utilised for statistical analyses. Data are presented as mean ± SD of at least three independent experiments.

## 5. Conclusions

Eight, and 48-week-old male F-344 Al-induced hypercholesterinaemic AD-like rats receiving Al and high cholesterol containing chow as well as healthy control animals underwent [^11^C]C-PIB PET examinations to assess Aβ-specific radiopharmaceutical uptake. Significantly increased SUV_mean_ values of the AD-resembling group were depicted compared to the normal cohort. One and two months after the reintroduction of normal diet, the cerebral [^11^C]C-PIB accumulation of AD-mimicking animals decreased by one half and a third, respectively, to the value measured after the administration of a month-long combined (Al_2_(SO_4_)_3_ and cholesterol) treatment. Therefore, we managed to establish a hypercholesterinaemic AD-resembling translational rat model that holds the potential to open a new era in the profound understanding of the neuropathology of AD enabling the investigation of novel therapeutic targets and radiopharmaceutical development at preclinical level.

## Figures and Tables

**Figure 1 ijms-23-13950-f001:**
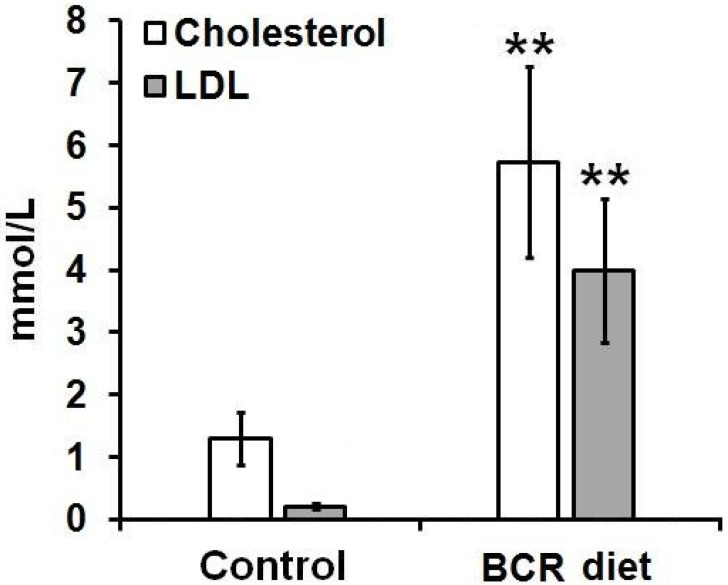
Effects of standard (Control) and BCR diets on serum lipid parameters (cholesterol and LDL) of the healthy control rats. Quantitative analysis of the serum cholesterol fractions after different feeding procedures. Significance level (compared to the control group): *p* ≤ 0.01 (**). BCR: butter and cholesterol rich; LDL: low-density lipoprotein.

**Figure 2 ijms-23-13950-f002:**
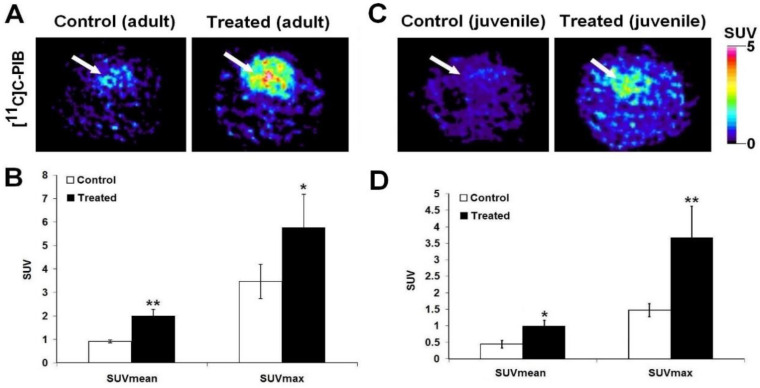
In vivo assessment of the impact of the administered combined (cholesterol and Al_2_(SO_4_)_3_) treatment on [^11^C]C-PIB brain radiopharmaceutical accumulation of both the older and the younger rats using [^11^C]C-PIB PET examinations. (**A**) Represents the transaxial PET slices of the untreated control (**A left**) and the treated (**A right**) adult group of rats. Panel B demonstrates the results of the quantitative SUV analysis of [^11^C]C-PIB uptake of the adult control rats, and those receiving treatment. (**C**) Represents the transaxial PET slices of the untreated control (**C left**) and the treated (**C right**) young rats. (**D**) Demonstrates the results of the quantitative SUV analysis of [^11^C]C-PIB uptake of the young control rats, and those receiving treatment. SUV values are presented as mean ± SD. Significance level (compared to the untreated control group): *p* ≤ 0.05 (*) and *p* ≤ 0.01 (**). [^11^C]C-PIB: [^11^C]C Pittsburgh compound B; PET: Positron Emission Tomography; SUV: standardised uptake value; SD: standard deviation. White arrow: brain.

**Figure 3 ijms-23-13950-f003:**
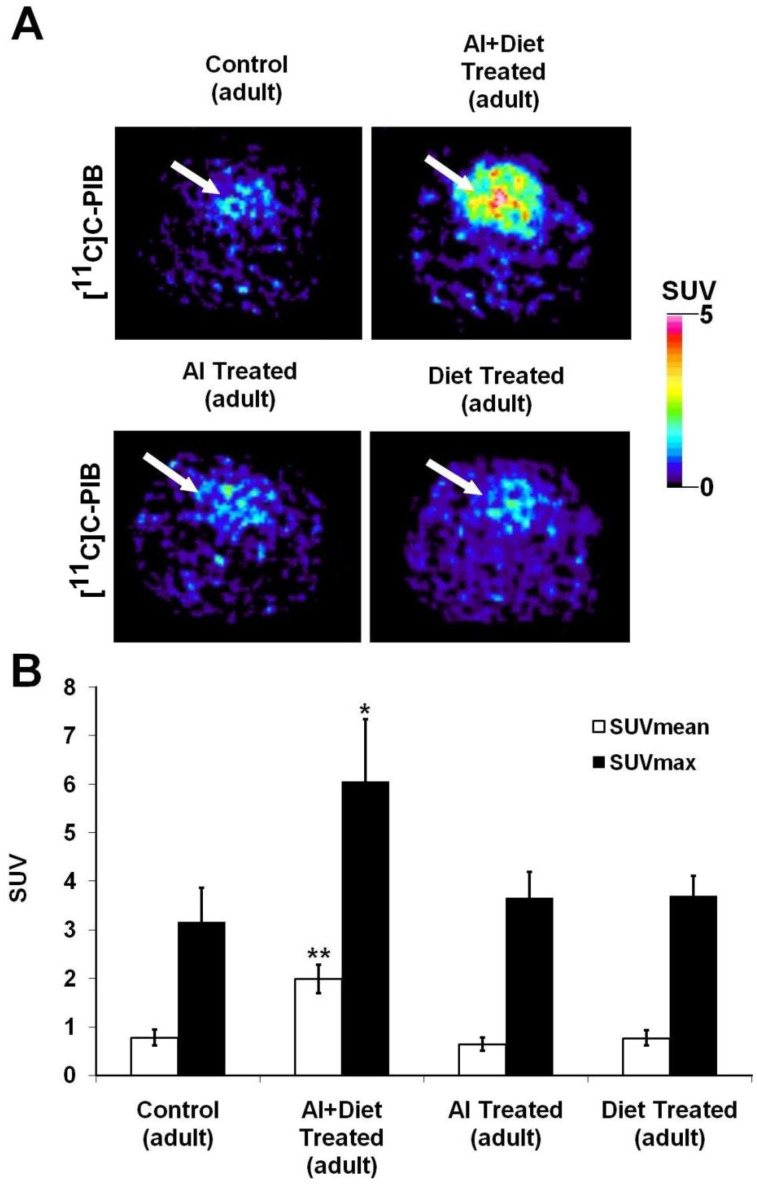
Comparison of cerebral [^11^C]C-PIB radiopharmaceutical uptake of the elderly animals administered with monotreatment (cholesterol or Al_2_(SO_4_)_3_) and combined (cholesterol and Al_2_(SO_4_)_3_) treatment to each other and to the control healthy group of old rats. (**A**) Shows the transaxial PET images of the old animals receiving standard chow and normal tap water (upper row, left), combined treatment (upper row, right), and Al-containing (lower row, left) or cholesterol-enriched (lower row, right) monotreatment. (**B**) Demonstrates the results of the quantitative SUV analysis of [^11^C]C-PIB uptake of the control, the monotreated rats and those receiving combined treatment. SUV values are presented as mean ± SD. Significance level (compared to the untreated control and monotherapy groups): *p* ≤ 0.05 (*) and *p* ≤ 0.01 (**). [^11^C]C-PIB: [^11^C]C Pittsburgh compound B; PET: Positron Emission Tomography; Al: aluminium; SUV: standardised uptake value; SD: standard deviation. White arrow: brain.

**Figure 4 ijms-23-13950-f004:**
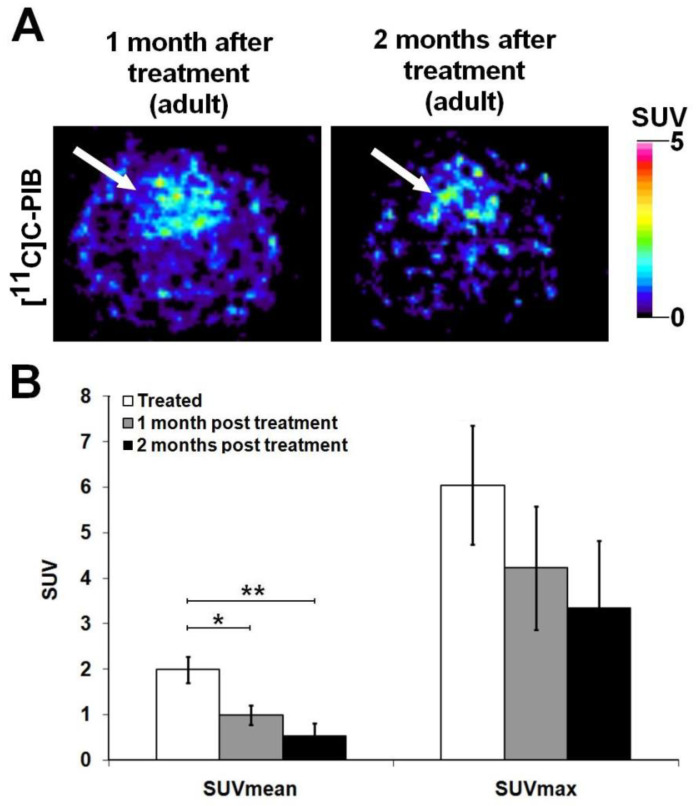
In vivo assessment of cerebral [^11^C]C-PIB accumulation one and two months after the restoration to normal diet and tap water. (**A**) Represents the transaxial PET slices of the rats previously receiving combined (cholesterol rich and Al_2_(SO_4_)_3_) diet one (**A left**) and two (**A right**) months after the restoration to normal rodent chow and tap water. (**B**) Presents the results of quantitative SUV analyses. One and two months after the reintroduction of normal diet, the cerebral [^11^C]C-PIB accumulation of AD-mimicking hypercholesterinaemic animals, decreased by half and third; respectively to the value measured after the administration of a month-long combined (Al_2_(SO_4_)_3_ and cholesterol) treatment. SUV values are presented as mean ± SD. Significance levels *p* ≤ 0.05 (*) and *p* ≤ 0.01 (**). PET: Positron Emission Tomography; [^11^C]C-PIB: [^11^C]C Pittsburgh compound B; SUV: standardised uptake value; SD: standard deviation. White arrow: brain.

**Figure 5 ijms-23-13950-f005:**
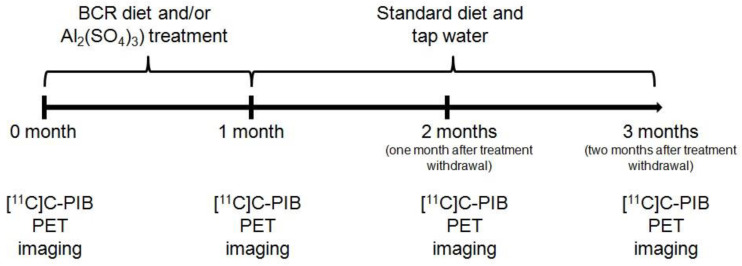
Study design. BCR: butter- and cholesterol-rich; PET: Positron Emission Tomography; [^11^C]C-PIB: [^11^C]C Pittsburgh compound B.

**Table 1 ijms-23-13950-t001:** Components of the standard VRF1 diet.

Standard VRF1 Diet	
Component	Composition (%)
moisture	10.00
crude protein	19.11
crude fat	4.75
crude fibre	3.85
ash	6.97
NFE	55.32

**Table 2 ijms-23-13950-t002:** Components of the special BCR diet.

BCR Diet	
Component	Composition (%)
Standard VRF1 feed	60.00
casein	9.50
butter	20.00
cholesterol	4.00
sodium cholate	1.00
“SM8” salt mixture, according to Sós [49]	5.00
6-Methyl-2-thio-uracil	0.50

BCR: butter and cholesterol rich.

## Data Availability

The dataset used and/or analyzed during the current study are available from the corresponding author on reasonable request.

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
