# Peer review of "In Vivo Preclinical Assessment of β-Amyloid–Affine [11C]C-PIB Accumulation in Aluminium-Induced Alzheimer’s Disease-Resembling Hypercholesterinaemic Rat Model"

_ijms, 2022, doi:10.3390/ijms232213950_

Round 1

Reviewer 1 Report

In the research area of Alzheimer’s disease (AD), transgenic rodents expressing the AD-associated gene such as amyloid precursor protein or tau are widely used as animal models of AD. Although these animal models are powerful tool for basic research of AD, one disadvantage of the models is that researchers always determine their genotype by using PCR assay, which is time-consuming and cost-ineffectiveness. In this manuscript (ijms-1959512), Képes and colleagues tried to generate a cost-effective rat model of AD. The authors chemically exposed rats to a combination of Aluminum ions and cholesterol, resulting in reproduction of amyloid beta (Ab) deposits, a major pathological hallmark of AD, in the brain. This Ab pathology seems to related to aging process because elder rats had a greater amount of the Ab deposits than young rats. Moreover, the Ab pathology was partially restored when the exposure of rats to the chemicals was stopped. Although it is interesting that the Ab pathology is reversible even in eldered rats treated with chemicals, the reviewer has some concerns and suggestions to improve the quality of the original manuscript.

Major points:

1.      One potential concern for this study is the association with whether the rat model of AD that the authors generated could exhibit AD-like clinical and pathological features other than the deposition of Ab. In other words, if the authors would like to conclude that the rats exposed to both Aluminum ions and cholesterol is a new animal model of AD, the authors should provide evidence for that rats exposed to the chemicals develop cognitive impairment, neuronal loss in the brain, activation of astrocytes and microglial, etc.

2.      Please mention the research question and/or the aim of the present study in Abstract. The reviewer did not understand the objective and intention of this work when read the Abstract.

3.      In Figure 5, the restoration of the Ab pathology even in elder rats after stopping the chemical exposure seems to be interesting. The readers of this paper would like to know molecular mechanism(s) underling how the Ab pathology is restored. Please provide evidence to explain the reversible phenomenon of the Ab pathology.

Minor points:

1.      Figure 2 should be combined with Figure 3 to help readers easily understand how the amounts of Ab are changed during the aging process.

2.      SUV is not listed in the section of Abbreviations.

3.      The paper by Dollken [Arch Exp Pathol Pharmacol (1897) 40 58] should be added to the list of Reference.

Reviewer 2 Report

The present study might develop an in vivo experimental model for Alzheimer's disease using rats. The authors utilized β-Amyloid-affine [11C]C-PIB, aluminum sulfate, and a high-cholesterol diet for developing the model. The quality of the study is good and more interesting to read the manuscript. However, I like to add some points for minor improvement of the present manuscript before publication.

1.     Line 20 in the abstract, “8 (young), - and 48-week-old (elderly)” is not correct. What is the number ‘8’ ?

2.     Line 20 in the abstract, “F-344 rats” can be changed to “Fischer-344 rats”.

3.     Line 111, what is the definition of “safe and rapid”?

4.     Line 124, “Figure 1. Effects of standard (Control) and BCR diets on healthy control rats”, the information is incomplete.

5.     Line 143 & 152. “applied combined”, not perfect.

6.     Line 223-234, the authors explained the use of rabbits and guinea pigs for the AD experiment model, and how this information is related to the present study objects. The objective of the study needs to explain appropriately.

7.     Line 239-245, the statements are not clearly explained. Particularly  “no remarkable elevation of brain cholesterol level was observed compared to the control animals”, this statement, how related to “achieve diet-based hypercholesterinaemia”.

8.     Line 299, again “8, - and 48-week-old male”, needs to be modified.

9.     The number of animals used in each group was only 4, how the authors selected the number of animals? Is the number sufficient to apply statistical significance?  Are the authors applied any power analysis calculation for finding the animal number for each group?

10.  Possible authors can add a figure, that explains the entire treatment and experimental time-line for easy understanding of experimental protocols.

11.  Line 362, “Eight, - and 48-week-old male”, needs to be modified.

12.  Finally, need to correct many typo errors, italic, and extra ‘-’.

Reviewer 3 Report

The authors in this article have managed to establish a hypercholesterinaemic AD-resembling translational rat model that holds the potential to open a new era in the profound understanding of the neuropathology of AD to enable identification of novel therapeutic targets at preclinical level. This is a novel work.

However, my minor concern is what would be the source of consumption of 200mg/L of Al2(SO4)3) in normal human being's daily routine. Tap water can't be a big source as water is purified/filtered using Al2(SO4)3) in most developing countries and their cognitive capability of the population there is still strong. Unless people start consuming water filtering products like Al2(SO4)3) in their daily diet specifically. 

Round 2

Reviewer 1 Report

In the revised manuscript (ijms-1959512), the authors have addressed all of my review comments with the exception of the additional experiments for the connection between Aluminum/cholesterol exposure and pathological/phenotypic features of Alzheimer’s disease. In the letter of the responses to me, the authors explained why they did not perform the additional experiments, and they showed the limitations and the future perspectives of this study. However, there is no such explanation in the revised manuscript. I strongly recommend that the authors should describe the limitations and the future perspectives of this study in the Discussion.
